# Humane and Comprehensive Management of Challenging Behaviour in Health and Social Care: Cross-Sectional Study Testing Newly Developed Instrument

**DOI:** 10.3390/healthcare11050753

**Published:** 2023-03-03

**Authors:** Sirpa Tölli, Raija Kontio, Pirjo Partanen, Anja Terkamo-Moisio, Arja Häggman-Laitila

**Affiliations:** 1Faculty of Health Sciences, Department of Nursing Science, University of Eastern Finland, FI-70210 Kuopio, Finland; 2School of Health and Social Care, Oulu University of Applied Science, FI-90101 Oulu, Finland; 3Department of Nursing Science, University of Turku, FI-20014 Turku, Finland

**Keywords:** restraint, trauma-informed care, human rights, scale development

## Abstract

Background: Management of challenging behaviour causes victimization and violates the human rights of service users in psychiatric and long-term settings for people having mental health issues and learning disabilities. The purpose of the research was to develop and test an instrument for measuring humane behaviour management (HCMCB). The research was guided by the following questions: (1) What is the structure and content of the Human and Comprehensive management of Challenging Behaviour (HCMCB) instrument, (2) What are the psychometric properties of the HCMCB instrument, and (3) How do Finnish health and social care professionals evaluate their humane and comprehensive management of challenging behaviour? Methods: A cross-sectional study design and STROBE checklist were applied. A convenience sample of health and social care professionals (n = 233) studying at the University of Applied Sciences (n = 13) was recruited. Results: The EFA revealed a 14-factor structure and included a total of 63 items. The Cronbach’s alpha values for factors varied from 0.535 to 0.939. The participants rated their individual competence higher than leadership and organizational culture. Conclusions: HCMCB is a useful tool for evaluating competencies, leadership, and organizational practices in the context of challenging behaviour. HCMCB should be further tested in various international contexts involving challenging behaviour with large samples and longitudinal design.

## 1. Introduction

Challenging behaviour occurs among people who are subjected to distressing conditions, experience excessive arousal, and/or have suboptimal cognition [1]. It is defined as behaviour that threatens the quality of life and/or physical safety of the individual or others, and these instances often lead to restrictive and aversive responses [2]. Challenging behaviour exhibits, for example, as self-harm, aggression, incoercible psycho-motor agitation, pervasive stereotypes, pica, and destruction of objects [3], and the feelings it invokes in others can be either intolerable or overwhelming [2]. It is common within vulnerable groups, for example, in mental health nursing [4], people with dementia [5], acquired brain injuries [6], intellectual disabilities [6,7], and in general nursing [8]. Furthermore, previous research has cited pain, stress, a lack of privacy, and long waiting times as underlying reasons for challenging behaviour among patients [9,10]. 

Challenging behaviour is often managed by restrictive interventions, which cause psychological and physical harm to patients [11,12]. The use of restraints and coercion has been acknowledged as a violation of patients’ human rights [12,13] and was reported to cause victimization [14]. World Health Organization argues that people with mental health conditions often experience severe human rights violations and discrimination [15] and reducing the use of restrictive interventions [11] and developing staff competence in humane mental health care services [16] are important international goals.

Multiple training interventions pursuing to enhance staff competence to manage challenging behaviour exist worldwide, yet previous research has not demonstrated a coherent or comprehensive impact on staff competence. Previous research into the competence needed to manage challenging behaviour has focused on single elements of competence using unilateral scales [1,4,10]. Variation among competence measurements hampers the evaluation of such interventions [4,10]. The Cochrane review [1] concluded that future studies should use the same well-established, validated questionnaires to produce comparable information when evaluating the management of challenging behaviour. 

Competent practitioners can replicate their performance to a satisfactory degree each time they use a specific measure [17]. This requires the correct knowledge, skills, and attitudes [17,18], along with psycho-social and psycho-motor elements [18] and professional confidence [19]. In the context of behaviour management, confidence is considered a key issue in the effectiveness of staff support [20], yet previous research has not provided evidence, nor definitions, for the optimal level of confidence needed for the management of challenging behaviour [10]. 

Further, a similar dilemma continues when measuring staff attitudes and knowledge in the context of behaviour management. Previous research has not provided clear evidence that training had an impact on staff attitude [4,21] or knowledge [10], yet they are considered an important element of competence. Previous research does not define “desired” attitude, although staff should be able to apply the Human Rights Act and other legal frameworks relevant to the use of restrictive interventions in a way that is compatible with the person’s rights [22]. In addition, patient safety has not previously been linked with competence [10].

Based on the research gap on the comprehensive competence to manage challenging behaviour and related patient safety issues in healthcare services, it is necessary to have valid and reliable yet generic instruments that enable assessment of nursing staff’s competence to manage challenging behaviour. A valid instrument could explain the relationships of staff competence with organizational factors protecting the human rights of service users.

The aim of this study was to develop and test an instrument for measuring the humane and comprehensive management of challenging behaviour. Our ultimate aim was to enable and enhance the ethically sustainable encountering of the patients belonging to the vulnerable group. The research was guided by the following questions:What is the structure and content of the Human and Comprehensive Management of Challenging Behaviour (HCMCB) instrument?What are the psychometric properties of the HCMCB instrument?How do Finnish health and social care professionals evaluate their humane and comprehensive management of challenging behaviour?

## 2. Materials and Methods

### 2.1. Research Design

A cross-sectional study including scale development process based on recommendations by [23]. All methods were carried out in accordance with relevant guidelines and regulations and are reported using the Strengthening the Reporting of Observational Studies in Epidemiology (STROBE) checklist (Appendix A) [24]. The instrument development included four phases (Figure 1).

### 2.2. Identification of the Need for the Instrument

The lack of a comprehensive instrument for measuring competence in managing challenging behavior was identified based on the results of two reviews. We conducted a complementary review (Figure 1) to our systematic review [10] across the same databases to identify possible newly developed instruments used in competence measurements, and we found 24 various instruments in the relevant published literature. All except one instrument measuring staff interaction were subjective self-reporting measures. (Table 1). 

Confidence was the most commonly measured competence, as it was investigated in five studies in the systematic review [10] and in ten studies in the complementary review (Table 1). Other competencies measured in the studies were attitudes, knowledge [10], quality of interactions, and empathy (Table 1). These reviews provided also the main concepts of the management of challenging behaviour. 

### 2.3. Item Generation 

An empirical approach was adopted to operationalize [23] the humane and comprehensive management of challenging behaviour. First, a qualitative method was applied to probe former psychiatric patients’ experiences of the behaviour management that staff utilized during their hospitalization; the former patients were also asked to describe which competencies psychiatric nursing staff need for behaviour management [12]. The study identified many reasons for challenging behaviour and various interventions used in behaviour management. The former patients identified delivering care based on patient needs to be a core competence for the staff. We used the results to generate the items concerning humanity, knowledge, and ethical sensitivity. 

Next, we used a qualitative method to explore the experiences of trained Management of Actual or Potential Aggression (MAPA) instructors (https://www.crisisprevention.com/en-GB/Our-Programs/MAPA-Management-of-Actual-or-Potential-Aggression (accessed on 25 February 2023)); these instructors deliver training and care for people who display challenging behaviour [25]. Based on the instructors’ perceptions, management is based on knowledge of the precipitating factors that may lead to challenging behaviour, with this knowledge including key legislation and ethical principles. For instance, staff should be able to form therapeutic relationships with people in distress to support their autonomy and protect the human rights of service users. Confidence in behaviour management was described as “the ability to support people in distress by applying the least restrictive verbal and/or physical measures in a compassionate way while maintaining self-awareness and self-control” [25]. (Figure 1) 

In addition to the reviews, the two qualitative studies provided the theoretical construct [23] of the instrument and an initial item pool (n = 155) was created within the research group. They further reviewed the items to detect overlaps yet ensure heterogeneity and reduced them in consensus. The first version of the instrument included 77 items using the VAS scale (1–10) and was named Humane and Comprehensive Management of Challenging Behaviour (HCMCB). The VAS scale was chosen due to its sensitivity in repeated measurements [23]. (Figure 1) 

### 2.4. Validation of the Instrument 

The content validity of the first version of the instrument was assessed in October 2020 by calculating the content validity index (CVI) [26,41] for individual items and the entire instrument in a one-round evaluation by a panel of nine experts. The panelists were chosen purposefully from various universities and research groups to ensure that they have professional insights for both caring for people displaying challenging behaviour, and scale development. The panelists were RNs or Bachelors of Social Services with experience in psychiatric nursing or rehabilitation in Finland. In addition, they all had either a Ph.D. degree or were Ph.D. candidates. To improve the instrument, the items that received CVI scores <0.66 were deleted (8 items). The ratings for clarity varied from 0.33 to 1. We did not exclude any items based on a low CVI score for clarity but instead amended the wording of these items to increase clarity.

The experts were also encouraged to provide written comments about the items, along with suggestions if they felt that certain relevant items were missing. Based on these suggestions, we added one item concerning documentation. In addition, panelists suggested formulating two grouped statements of existing items of Knowledge and Organizational culture and scoring them using a five-point Likert scale to increase the speed at which the scale could be completed. They both included 10 items. This modified, second version of the instrument included 70 items. (Figure 1)

The face validity of the second version of HCMCB, in electronic format, was tested in November 2020 on a voluntary group of 23 Finnish MAPA instructors working in health and social care with people who display challenging behaviours. The participants were recruited via an instructor’s Facebook community. The community included 160 instructors, and they all were invited to participate. A participant had to be a professional working in health and social care or in education with completed MAPA Instructor training. The participating instructors were asked to complete the questionnaire and record how long it took them to answer. The instructors were also asked to assess the comprehensibility, clarity, and length of the instrument using open-ended questions. A total of 17 written comments were received and carefully analyzed, and the mean time for completing the instrument was 12 min. Feedback from the instructors demonstrated that the instrument had adequate structure and an acceptable number of items. Some of the comments highlighted that it was difficult to answer items related to pharmaceutical care and physical restraint, as not all of the participants had an experience with this type of care; moreover, physical restraints are not used in all workplaces. Therefore, the two items related to pharmaceutical care and the seven items related to physical restraint were grouped together as optional items, only to be completed if they were relevant to the respondent’s work. Moreover, one item related to the support that staff needs when dealing with the emotional burden caused by challenging behavior was added and placed onto grouped statements of Organizational culture using a five-point Likert scale. This third version of HCMCB (Table 2) included 71 items and was administered for further evaluation.

### 2.5. Testing the New Instrument 

#### 2.5.1. Participants and Research Context

A convenience sample of health and social care professionals was recruited. We decided to recruit our participants from universities of applied sciences (n = 13) due the COVID-19 pandemic has been shown to decrease the nursing staff’s willingness to participate in research projects. In addition, Finnish universities recommended not recruiting frontline staff working in clinical environments for research purposes due to their increased workload. 

In Finnish universities of applied sciences, applicants must have a minimum of two years of clinical work experience to be eligible to apply for a Masters’ degree program in health and social care. We assumed that Masters’ degree students had experiences of patient challenging behaviour, and they might be willing to participate due to their ongoing academic studies. Selecting Masters’ degree students representing various professional groups as participants were assumed to ensure a sample size of five to ten subjects per item for conducting Exploratory Factor Analysis (EFA) [23,42] and enable the assessment of the instrument’s genericity. Therefore, a participant had to be a health or social care professional (e.g., Registered Nurse, Occupational Therapist, or Bachelor of Social Services) with a Bachelor’s degree who was studying in a Masters’ degree Program at a university of applied sciences. All of the students who were enrolled in a Masters’ degree program (n = 1685) at universities of applied sciences were invited to participate in the study. The participants were invited via email by a contact person at each university. A total of 233 students completed the questionnaire.

#### 2.5.2. Data Collection and the Instrument 

Data were collected from December 2020–March 2021 electronically through Webropol software (V3.0; Webropol Oy, Helsinki, Finland). The Humane and Comprehensive Management of Challenging Behaviour (HCMCB) instrument consisted of four sub-scales: Preventing and facing challenging behavior; Managing Challenging behavior; Teamwork; and Organizational culture (Table 2). The instrument included seven theoretical dimensions and 71 positively-worded items that respondents were asked to score using a five-point Likert scale and a ten-point Visual analogue scale (VAS) ranging from totally disagree to totally agree (Table 2). Items were graded in the same direction and they were allocated and mixed in four sub-scales. This was conducted because we did not want the titles of theoretical dimensions, for example, ‘Attitude’, to influence participants’ thoughts. The theoretical dimensions were: knowledge (10 items, five-point Likert scale); skills (12 items, VAS 1–10); attitude (seven items, VAS 1–10); confidence (five items, VAS 1–10); ethical sensitivity (11 items, VAS 1–10); teamwork (six items, VAS 1–10); and organizational culture (20 items, 11 items scored using a five-point Likert scale and nine items scored using a VAS 1–10). The completed instrument may include 62, 64, 69, or 71 items based on the number of optional items that the participant answers (these depend on their experience with pharmaceutical care and physical restraint).

The HCMCB also measured certain background variables (15 items), namely, gender, age (years), academic level of education, type of professional education, work experience (years), area of work (health or social care), work environment (hospital, office, home), age group of service users, the status of employment (permanent or temporary), employer (public, private, third sector), behavior management training in primary professional education, continuous training in behavior management, the frequency at which verbal interventions were used to manage challenging behaviors in current work, and the frequency at which physical interventions were used to manage challenging behaviors in current work.

#### 2.5.3. Data Analysis

SPSS (version 27.0, SPSS Inc., Chicago, IL, USA) was used to analyze the data. Principal Component Analysis demonstrated high intercorrelations between items on both scoring scales. Thereafter, EFA (principal axis factoring with Varimax rotation and an eigenvalue of 1) was applied to individual theoretical dimensions to discover the explanatory factors. The Kaiser–Meyer–Olkin (KMO) measure was employed to verify sampling adequacy. Cronbach’s alpha values were calculated for all of the resulting factors to evaluate internal consistency [42,43]. All of the calculated Cronbach’s alpha values were interpreted using suggestions by [43]. Finally, sum variables were calculated for each distinct dimension to investigate the participants’ humane and comprehensive management of challenging behavior. All items were set as mandatory in the Webropol survey for completing the scale, which is why there is no missing data as such. Variations in the number of respondents regarding the optional items of pharmaceutical care and physical restraint were ignored automatically during the SPSS analysis.

#### 2.5.4. Ethical Considerations

Permission to perform this study was obtained from thirteen Finnish universities of applied sciences. Participants were provided with information about the study via email, including an explanation that their participation was voluntary. Participation or withdrawal was not controlled by universities or the researcher and had no influence on students’ ongoing studies or study assessments. Participants were required to give informed consent at the beginning of the electronic survey; more specifically, they had to choose “agree” on a mandatory item about willingness to participate. No identifiable information was collected from participants.

## 3. Results

### 3.1. The Structure and Content of the Tested Version of HCMCB

Completing the questionnaire took approximately 13 min (range 6–22). Based on the results of EFA, the original theoretical dimension of Organizational culture was divided into two, namely Organizational Culture and Leadership. Moreover, the tested version of HCMCB comprised eight dimensions (knowledge, skills, attitude, confidence, ethical sensitivity, teamwork, organizational culture, and leadership), demonstrated a 14-factor structure, and included a total of 63 items (Table 3). 

A total of eight items were removed based on the EFA results, i.e., these items either showed weak correlations or did not load to any factor. Of the removed items, two belonged to the Knowledge dimension (Other service users and restrictive environment may cause challenging behavior), one belonged to the Attitude dimension (I try to avoid the physical restraint if possible), four belonged to the Ethical sensitivity dimension (I evaluate the suitability of the restrictive measures from the perspective of the service user; I conduct physical restraints only when the service user injures themselves or others’; Physical restraint may be the most humane way to support a service user displaying challenging behavior; We evaluate the ethicality of restrictive practices in my unit), and one belonged to the Organizational culture dimension measuring patient safety (Physical health is evaluated from the service users that are subjected to physical restraints) (Table 3).

### 3.2. Psychometric Properties of Tested HCCMB

The KMO values ranged from 0.651 (Knowledge) to 0.888 (Leadership), while the eigenvalues ranged from 1.105 (Person-centered care) to 5.449 (Supporting service-users’ self-control). The calculated Cronbach’s alpha values ranged from 0.535 (for three items of ‘Best interest’) to 0.939 (for two items of ‘Debriefing’), while commonalities ranged from 0.209 (My organization offers supervision to deal with the emotional stress caused by service users’ challenging behavior) to 0.790 (We conduct a debriefing conversation with the service user after every restrictive incident). The percentage of variance explained by the individual factors ranged from 58.866 (Self-control when restraining) to 11.511 (Safety management) (Table 3).

### 3.3. Humane and Comprehensive Management of Challenging Behaviour among Finnish Health and Social Care Professionals

The study participants were between 25 to 61 years of age, with 90% women. Most of the participants (80%) were healthcare professionals, and they had between two to 35 years of work experience. Most of the participants (32.2%) worked in clinical nursing, while elderly care (16.3%) and psychiatric nursing or substance abuse work (15.1%) were also common professional contexts among the participants. Furthermore, 35% of the participants reported that their professional education had included behaviour management training, and almost 60% had had such training during their continuous education. Almost 80% of the participants reported that challenging behaviour occurred at least once per month, while 55% of the participants reported using verbal skills to manage challenging behaviour at least once a week, and 31% of the participants reported using physical skills in behaviour management at least once per month. (Table 4.)

Half of the participants (52%) conduct physical restraints in their work, and two-thirds (72%) of the participants participate in pharmaceutical care. The participants’ evaluations of their humane and comprehensive management of challenging behaviour ranged from 4.5 to 8.8 in dimensions rated using a VAS scale (1–10) and from 2.7 to 4 in dimensions rated using a five-point Likert scale (Table 2). Knowledge of self-regulation demonstrated a mean score of 4 (SD 0.7), Attunement skill showed a mean score of 8.15 (SD 1.13), and Supporting service-users’ self-control demonstrated a mean score of 7 (SD 1.72). Humanity had a mean score of 8.7 (SD 1.29), while Person-centered care demonstrated a lower mean score (5.47; SD 2.49). Self-control when restraining demonstrated a mean score of 8.85 (SD 1.37). Furthermore, ‘Best interest’ showed a mean score of 8.85 (SD1.37), clarity of values demonstrated a mean score of 7.76 (SD 2.10), fluent teamwork had a mean score of 8.35 (SD 1.83), and debriefing demonstrated a mean score of 4.53 (SD 3.28). Competence management received a mean score of 2.75 (SD 1.11), while safety management had a mean score of 2.77 (SD 1.13). Finally, restraint reduction showed a mean score of 4.94 (SD 2.58), while service users’ safety demonstrated a mean score of 7.75 (SD 1.65). We did not examine subgroups due to the heterogeneity of participants’ professional backgrounds and low response rates. 

## 4. Discussion

### 4.1. Consideration of the Content and Structure of the HCMCB Instrument

This study respected the procedures advocated in the study by [44]. To the best of the authors’ knowledge, HCMCB is the first instrument that was designed to capture the humane and comprehensive management of challenging behaviour. Previous research has reported the unilateral measurement of individual competencies; little attention has been paid, for example, to empathy and interaction, although they are crucial in the management of challenging behaviour [12,25]. Furthermore, previous research has not been able to comprehensively describe competence or confidence related to behaviour management [10]. Our instrument covers the competencies included in previous measures [10], and we utilized the novel conceptual formulations of studies [12,25] when designing the HCMCB [23]. In addition to individual competencies, we wanted to measure teamwork and patient safety linked to the organizational context.

HCMCB highlights compassion and the humane perspective in behaviour management within health and social care by also capturing the ethical perspective. We believe that it manifests how organizations’ practice protects human rights and the safety of vulnerable people, which have often been violated in psychiatric care [12,13,14,45]. The presented instrument can be used to develop new training courses, as well as evaluate established behaviour management interventions. These indirect effects will support the efforts to minimize coercive practices and the implementation of new training standards by the Restraint Reduction Network [22], which aims to ensure that healthcare professionals receive high-quality behaviour management training, service users’ human rights are protected, and restrictive practices are minimized. The structure of the HCMCB enables the recognition of organizational factors, such as leadership, related to the ethical management of challenging behaviour. The instrument provides knowledge that can be used to develop organizational and leadership practices that enhance the management of challenging behaviour in clinical work [46]. Hereby, the recognition of the context is important for the successful implementation of new human practices when managing challenging behaviour among vulnerable people. 

### 4.2. Consideration of the Psychometric Properties of the HCMCB Instrument

The reliability of HCMCB, as well as the clarity and relevance of individual items, was assessed by a panel of nine experts, which exceeds the recommended expert panel size of seven [26]. The face validity, which describes suitability and comprehensibility, was tested with MAPA instructors, who are familiar with the measured phenomena and the vocabulary. This was important to confirm that the instrument measures what it was designed to measure [23]. 

Following EFA, the tested instrument included a total of 63 items over eight dimensions and 14 factors. According to the calculated Cronbach’s alpha values, the reliability of the instrument varied from poor to excellent. Most factors demonstrated good internal consistency, while two factors in the ‘Attitude’ dimension demonstrated questionable internal consistency, and the factor ‘Best interest’ in the ‘Ethical sensitivity’ dimension demonstrated poor internal consistency [43]. As the KMO values were appropriate (<0.5), demonstrating a robust factor structure, a low Cronbach’s alpha value may be explained by poor interrelatedness between items or heterogeneous constructs, both of which signal a need to revise or discard some of the items [27]. However, Cronbach’s alpha value may also underestimate reliability if computed with multidimensional items serving as the unit of analysis; this is because the items are not true-score equivalent [47]. Due to the optional response choices regarding items of pharmaceutical care and physical restraint, three factors presented smaller n than other factors. 

The VAS scale was chosen in the first version due to its sensitivity in repeated measurements [23]. The expert panel suggested using a five-point Likert scale in the grouped statements, and we followed the suggestion in the tested version of the instrument. The suggested change in the scaling may increase the user-friendliness of the instrument, yet mixing the scaling hampers the reliability and validity of testing and scoring, which should be considered in further development. 

More research among health and social care professionals caring for people displaying challenging behaviour will be necessary to demonstrate the reliability of HCMCB. Suitable clinical contexts for this research could be mental health nursing, dementia, and learning disability care. We suggest that the eight items that were removed based on the EFA results should be included in future evaluations of HCMCB due to their possible clinical relevance when ensuring a humane nursing style and the least restrictive practice. Coercive practice is connected to staff attitudes and ethical beliefs, and a staff member may lack the moral courage to change an existing practice [48]. In addition to reliability, the validity [44] of HCMCB needs further testing. The instrument also should be officially translated into the English language and tested in the international context. 

### 4.3. The Humane and Comprehensive Behaviour Management of Finnish Professionals

The prevalence of challenging behaviour in Finnish health and social care in this study was similar to what has been reported before in the study by [49], where 94.1% of the participants in the survey had experienced verbal abuse and 69.8% had experienced physical aggression in the previous twelve months. Despite the generality of challenging behaviour in the nursing profession, only 35% of participants reported that behaviour management training was included in their professional education. Behaviour management training is not systematically included in undergraduate nurses’ curricula [50], an issue that should be addressed in the future. Nurses have an important responsibility to recognize and protect the rights of people with mental health issues [51], and the competence to safeguard patients when using restraints should be prioritised in their education [45]. 

Participants reported rather positive evaluations of their individual competencies and were most critical in their evaluations of ‘Person-centered care’, ‘Debriefing’, ‘Competence and safety management’ and ‘Restraint reduction’. These findings suggest that Finnish health and social care organizations should review their policies regarding the use of restraints, along with the prevalence of restrictive practices. Previous research suggests that the development of competence to manage challenging behaviour is influenced by leadership and organizational culture [25]. Strong organizational commitment is crucial to reducing violence [51,52,53], and the implementation of interventions that aim to reduce restraint techniques requires commitment from senior leaders [54,55], as the managers of any organization are expected to lead by example [56]. Leadership approaches to managing challenging behaviour and coercive measures should be complemented and calibrated by an emphasis on relational approaches to care [57], in which the HCMCB instrument might be useful. 

However, it should be noted that we did not use any institutional indicators, for example, the incidence of violence or restraint, that were representative of the participating organizations to verify the evaluations of competence as suggested in the Cochrane review by [1]. Therefore, we cannot make conclusions about the participants’ competence or the quality of care in participants’ organizations. This needs to be considered in future evaluations and review relevant incident reports of the use of restraints alongside questionnaire results. In addition, service users’ feedback provides important knowledge of the experienced quality of care. Further studies are needed to examine how self-reported competence to manage challenging behaviour translates to objectively assessed competence in practice and how this may, in turn, influence safety and the protection of the human rights of people that are subjected to the use of restraints. In addition, the competence was measured only once instead of repeated measurements, and no conclusions of the exposure for change of the competence can be made. This requires longitudinal design in future research. However, the results indicate how to develop interventions applied in behavior management. 

### 4.4. Limitations 

The presented research included several limitations. For instance, the participating professionals may have a better conceptual understanding than frontline clinical staff due to their academic studies; hence, it may be necessary to revise certain items to improve the reliability of the scale. 

Moreover, the research showed a low response rate; for this reason, the presented results are not generalizable. One thousand six hundred eighty-five students were invited to participate, yet authors do not know how many of the invited students were active students following universities’ emails. The questionnaire was initially opened by 576 students, with 373 students starting the survey and 233 students completing the scale. This may indicate that the topic was not relevant [24] to all of the participants. The researcher (S.T.) received feedback from six students via email during the data collection; in these emails, the students expressed that they had experienced difficulties in completing the scale because it was not applicable to their work. It is possible that the participants’ work differed from the work of frontline staff in terms of the prevalence of challenging behavior, organizational practices regarding the use of restraints, or the nature of their work. Responding to all items was mandatory for completing the scale, and there was no “not relevant in my work” choice which could have increased the response rate. 

On the other hand, also the current pandemic situation with increased workload may have influenced the participation as most of the Masters’ degree students are known to work alongside their studies. The researcher was in contact with the contact persons from various organizations several times to ensure successful data collection. Since the researcher could not contact students personally, the invitations and information delivery depended on the contact person in each organization. We exceeded the minimum required sample size ratio for EFA (at least three participants per item) suggested by [58], but a larger sample could have improved the generalizability of the conclusions [24]. The use of two measurement scales complicates the investigation of the reliability and validity of the instrument, as well as the integration of the scores. Therefore, altering the scoring into one scale needs further consideration. The reliability and validity of the instrument need more testing, for example, assessing the sensitivity to change and stability with a test-retest method. 

## 5. Conclusions

This study developed and tested a novel instrument for measuring the comprehensive management of challenging behaviour (HCMCB), which proved to be valid and useful among staff working with people who display challenging behaviours. The psychometric properties of the instrument were satisfactory; thus, the developed instrument can be utilized to explain the relationships of staff competence with organizational practices related to the use of restraints. However, in further development of the instrument, the limitations need to be addressed, in particular concerning the scaling of the items. The instrument should also be tested further with bigger samples and the test–retest method in mental health nursing and other clinical contexts among staff working with individuals who display challenging behaviour to provide more evidence of its reliability, validity, and generalizability to health and social care. HCMCB ratings need to be interpreted with other institutional indicators, such as incidence reports of restraint use, to provide a reliable conclusion of the actual management of challenging behaviour.

## Figures and Tables

**Figure 1 healthcare-11-00753-f001:**
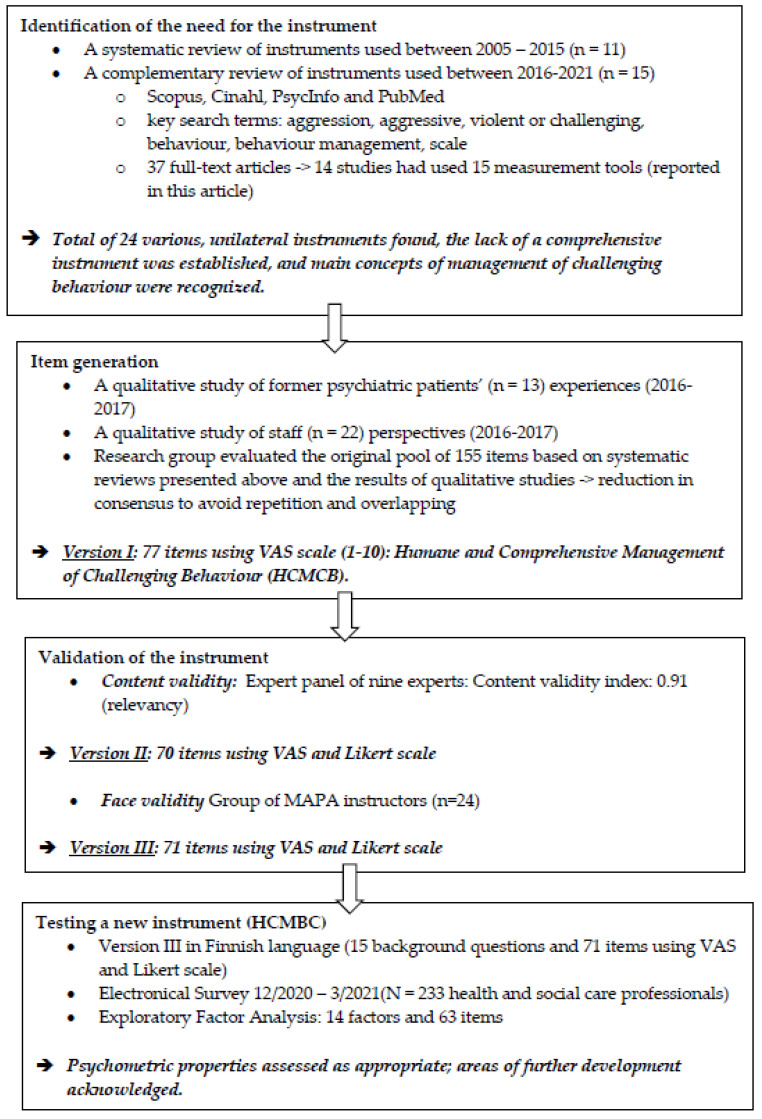
Phases in the development of the HCMCB instrument [10,12,23,25,26].

**Table 1 healthcare-11-00753-t001:** Instruments used between 2016–2021 to measure competence in managing challenging behavior.

Competence	Instruments, Items, and Clinical Context	Validation	Operationalization	References
Attitude	The Controllability Beliefs Scale (CBS); 15 items, five-point Likert scale Intellectual disability care.	Yes	The carer’s values and approaches towards people with learning disabilities.	[27]
	Approaches to Dementia Questionnaire (ADQ); 19 statements, five-point Likert scale Dementia care.	Yes	The respondent’s attitudes towards people with dementia	[28]
	MAVAS; 27 items, visual analogue scale (VAS) with a 100-mm line Various clinical settings in a hospital	Yes	Attitudes towards causes (internal, external and situational) and management of aggressive incidents	[29]
Knowlede	ASD (Autism Spectrum Disorder) questionnaire; 11 items. Actual changes were measured with a five-item questionnaire. Learning disability care	No	Perceived knowledge and actual changes in the knowledge.	[30]
	ID (Intellectual Disabilities) questionnaire; 10 items. Actual changes were measured with five-item questionnaire. Learning disability care	No	Perceived knowledge and actual changes in the knowledge.	[30,31]
	The Challenging Behaviors Attributions Scale (CHABA); 33 items Learning disability care Forensic mental health setting	Yes	A staff member’s explanation for challenging behavior: biomedical; emotional; learned (positive and negative); physical environment; stimulation	[31]
Confidence and coping	Causal Dimension Scale-II (CDS-II), adapted observer causal attributions; 12 items, nine-point scale Forensic mental health setting	Not reported	How staff members attributed the locus of causality, external control, personal control, and stability.	[32]
	Confidence in Coping with Patient Aggression Instrument (adapted version); 8 items, seven-point Likert scale. Forensic mental health setting	Not reported	The participant’s self-reported confidence in effectively and safely managing challenging behavior using the PBS method.	[33]
	Ways of Coping Questionnaire Interviews (WOCQ) (Swedish version); 40 items, four-point Likert scale Learning disability care	Yes	The carer’s use of coping strategies (emotion- and problem-focused coping skills).	[34]
	Challenging Behavior Self-Efficacy (CBSE); five items, seven-point scale Learning disability care	Not reported	Staff-perceived self-efficacy in relation to challenging behaviors.	[33,35,36]
	Sense of Competence in Dementia Care Staff (SCIDS) (Japanese version); 17 items, four-point Likert scale. Dementia care.	Yes	Competence in dementia care across four subscales: professionalism; building relationships; care challenges; and sustaining personhood	[28,37]
	A four-item questionnaire; Four items, 11-point Likert scale Learning disability care	Not reported	The participant’s self-efficacy in understanding the causes of challenging behavior and developing plans to reduce this type of behavior	[38]
	Self-reported confidence in aggression management skills; Six items, five-point Likert scale Paediatric hospital	Not reported	Self-reported confidence in managing clinical aggression (Being a group leader, verbal de-escalation, maintaining patient safety, hands-on and hands-off restraint, chemical restraint)	[39]
Interaction	Observation system based on self-determination theory (SDT) (modified); Three seven-point rating scales for staff behavior. Learning disability care.	Not reported	The three basic psychological needs of self-determination theory: autonomy; relatedness; and competence.	[40]
Empathy	Staff Empathy for people with Challenging Behavior Questionnaire (SECBQ); Five items, six-point agreement scales Learning disability care.	Yes	Five items related to understanding the behaviour of people with learning disabilities.	[33]

**Table 2 healthcare-11-00753-t002:** Structure of the version III of HCMCB.

Sub-Scale	Content and Scale	Remarks
Background questions (n = 15)		
Preventing and facing challenging behavior (n = 24)	knowledge (n = 10) Likert skills (n = 10) VAS attitude (n = 3) VAS confidence (n = 1) VAS	Two optional items (14 and 15) related to medication: these are chosen if medical treatment is part of the participant’s work.
Managing challenging behavior (n = 17)	skills (n = 2) VAS attitude (n = 3) VAS confidence (n = 4) VAS ethical sensitivity (n = 6) VAS organizational culture (n = 2) VAS	Seven optional items (26–32) related to physical restraint: these are chosen if conducting physical restraints is part of the participant’s work.
Teamwork (n = 7)	teamwork (n = 6) VAS attitude (n = 1) VAS	
Organizational culture (n = 23)	culture (n = 18) Likert and VAS ethical sensitivity (n = 5) VAS	

**Table 3 healthcare-11-00753-t003:** Structure, content, and psychometric properties of validated HCMCB instrument.

Dimension/Factor/Item/Scale	n *	Min	Max	Mean	SD	Loading	α **	KMO	Communalities	Eigen- Values	% of Variance
KNOWLEDGE, Likert								0.651			
Knowledge of self-regulation	233	1	5	4	0.7		0.81			3.543	44.289
-Limited potential for influence one’s treatment						0.57			0.288		
-Challenges in self-expression						0.538			0.448		
-Challenges in self-regulation						0.622			0.457		
-Stress						0.711			0.421		
-Uncertainty of one’s situation						0.54			0.318		
-Frustration or disappointment						0.652			0.399		
-Re-traumatization						0.579			0.316		
-Not obtaining the needed support						0.6			0.389		
SKILLS, VAS								0.851			
Attunement skill	233	3	10	8.15	1.13		0.735			1.872	13.37
-I recognize the early signs when challenging behaviour begins						0.336			0.342		
-I recognize my emotional reactions in interaction with service user						0.442			0.4		
-I recognize if my behaviour provokes service users						0.484			0.462		
Supporting service users’ self-control	169	2	10	7	1.73		0.829			5.449	38.923
-I encourage service users to discuss the reasons causing challenging behaviour						0.715			0.561		
-I offer information to service users about the factors regarding their challenging behaviour						0.787			0.581		
-I discuss with service users the suitable methods to calm themselves						0.737			0.578		
-I use augmentative and alternative communication when needed						0.356			0.226		
-I use violence risk assessment tools						0.525			0.364		
-I evaluate the adequacy of psychotropic medication based on service users’ feedback and behaviour						0.596			0.593		
-I recognize the symptoms of psychiatric illnesses from side-effects of medication						0.527			0.376		
-I verbally calm escalating behaviour						0.541			0.408		
-I give clear instructions for self-calming when challenging behaviour begins						0.688			0.51		
ATTITUDE, VAS								0.677			
Humanity	233	2	10	8.7	1.29		0.659			3.028	43.258
-I treat service users as “human to human”						0.793			0.469		
-I show service users that I understand their feelings and experiences						0.787			0.452		
-I always use the least restrictive method when restricting challenging behaviour						0.343			0.173		
Person-centered care	233	0	10	5.47	2.49		0.611			1.105	15.781
-I follow the individual plan that was made with service user for managing challenging behaviour						0.593			0.24		
-I implement various calming methods to avoid using restraints						0.655			0.269		
-I utilize the information gained from debriefing to find the best solution for future challenging situations						0.46			0.17		
CONFIDENCE, VAS								0.772			
Self-control when restraining	121	2	10	8.85	1.37		0.803			2.943	58.866
-I can control my behaviour if I get provoked by service user’s behaviour						0.632			0.411		
-I retain my self-control during restraint						0.88			0.63		
-I can calm myself after challenging situation						0.672			0.442		
-I am not afraid to conduct physical restraint if needed						0.574			0.33		
-I observe service users’ vital signs during physical restraint						0.72			0.453		
ETHICAL SENSITIVITY, VAS								0.752			
Clarity of values	233	1	10	7.76	2.1		0.8			2.924	36.545
-The restrictions used in my unit do not cause conflicts among staff						0.643			0.473		
-My personal values do not conflict with the restrictions applied in my unit						0.717			0.547		
-Staff in my unit do not use restrictive measures to make their work easier						0.782			0.324		
-Staff in my unit do not make threats of unnecessary use of restraints						0.655			0.281		
Best interest	233	2	10	8.85	1.37		0.535			1.359	16.989
-It may be a neglection if I choose not to respond service user’s challenging behaviour						0.477			0.271		
-There is nothing to hide with the restrictions I use, and they are ethically acceptable						0.403			0.321		
-I protect the privacy of service user if their behaviour compromises social norms						0.702			0.194		
TEAMWORK, VAS								0.725			
Fluent teamwork	233	1	10	8.35	1.83		0.806			3.173	52.883
-I inform my team about service user’s challenging behaviour to be prepared for possible risks						0.721			0.464		
-I recognize when my team needs me in the restraint						0.81			0.52		
-I evaluate the restraints required by the service user with my team						0.795			0.503		
-All team members know their role and act accordingly during restraint						0.656			0.434		
Debriefing	233	0	10	4.53	3.28		0.939			1.428	23.801
-We conduct debriefing conversations with service user after every restrictive incident						0.931			0.79		
-We evaluate the best time to conduct the debriefing from service user’s perspective						0.916			0.786		
LEADERSHIP, Likert								0.888			
Competence management	233	1	5	2.75	1.11		0.875			5.428	49.345
-Staff competence to manage challenging behaviour is regularly evaluated						0.709			0.632		
-Staff members’ physical health regarding conducting physical restraints is regularly evaluated						0.647			0.586		
-Staff is trained to manage challenging behaviour						0.742			0.574		
-Staff has common understanding of what is meant by restrictions						0.688			0.539		
-Restrictions are clearly defined in my unit						0.802			0.702		
-We have clear instructions for documentation of restrictive incidents						0.781			0.689		
-My organization offers supervision to deal with the emotional stress caused by service users’ challenging behaviour						0.345			0.209		
Safety management	233	1	5	2.77	1.13		0.753			1.266	11.511
-My unit has appropriate physical environment to care for service users’ displaying challenging behaviour						0.538			0.414		
-We have enough staff to manage challenging behaviour safely						0.852			0.474		
-We quickly obtain additional help from another unit in difficult situations						0.582			0.325		
-The measures applied during physical restraints in my unit are evidence-based.						0.32			0.593		
ORGANIZATIONAL CULTURE, VAS								0.801			
Restraint reduction	233	0	10	4.94	2.58		0.815			3.674	40.821
-Management in our organization supports the restraint reduction						0.735			0.548		
-Ward manager offers concrete examples of means to reduce the use of restraint based on organizational recommendations						0.852			0.619		
-The risk for violence is evaluated multi-professionally						0.625			0.468		
-The service user feedback is utilized in the development of the behaviour management						0.667			0.43		
Service users’ safety	121	3	10	7.75	1.65		0.647			1.278	14.2
-Physical restraints used in my unit do not cause physical harm to service users						0.475			0.239		
-Staff in my unit understand the difference between ‘support’ and ‘restraint’						0.542			0.306		
-Restrictions applied in my unit cause only temporary emotional harm to service users						0.644			0.253		
-Prevention of challenging behaviour has reduced the situations compromising safety in my unit						0.368			0.494		

* Variation in the number of respondents depends on their participation in pharmaceutical care and/or physical restraints in their workplace. ** Cronbach’s Alpha.

**Table 4 healthcare-11-00753-t004:** Characteristics of the sample (n = 233).

		n%	
Age (years) (Mean = 38.95 SD = 8.065)	25–34 35–44 45–54 55 and above	85 82 56 10	36.5 35.2 24 4.3
Gender	Male Female	24 209	10.3 89.7
Education	Health care Social care Rehabilitation Education	186 32 12 3	79.8 13.7 5.2 1.3
Highest academic degree	Scientific university Masters’ Degree in university of applied sciences Bachelor’s Degree in university of applied sciences	2 84 231	0.9 36.1 95
Current work	Clinical nursing Elderly care Psychiatry and substance abuse work Learning Disability nursing Children services Education Other	75 38 35 17 10 5 53	32.2 16.3 15.1 7.3 4.5 2.1 22.7
Employer	Public Private Third sector	185 45 3	79.4 19.3 1.3
Job status	Permanent Fixed term On call	207 24 2	88.8 10.3 0.9
Work environment	Hospital ward Residential home Reception (office) Emergency care Service users’ home Foster care Day center Daycare for children School Something else	52 38 36 20 17 6 4 4 3 53	22.3 16.3 15.5 8.6 7.3 2.6 1.7 1.7 1.3 22.7
Experience in initial profession	[mean = 12,24; SD = 7.164]		
My professional education included behaviour management training	Yes No Can’t tell	81 123 29	34.8 52.8 12.4
I have had behaviour management training as continuum training	Yes No	139 94	59.7 40.3
I face challenging behaviour in my work	Never Less than once in 6 months Once a month Weekly/daily	4 46 68 115	1.7 19.7 29.2 49.4
I use verbal skills in behaviour management in my work	Never Less than once in 6 months Once a month Weekly/daily	4 33 69 127	1.7 14.2 29.6 54.5
I use physical skills in behaviour management in my work	Never Less than once in 6 months Once a month Weekly/daily	86 75 35 37	36.9 32.2 15 15.9

## Data Availability

Data are not shared due to ethical issues.

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
