# Peer review of "Humane and Comprehensive Management of Challenging Behaviour in Health and Social Care: Cross-Sectional Study Testing Newly Developed Instrument"

_healthcare, 2023, doi:10.3390/healthcare11050753_

Round 1

Reviewer 1 Report

Dear authors:

Thank you for allowing me to read your work.

Although it is an interesting article, it is very extense and very confuse to read.

Some comments:

- Abstract should clearly state: introduction, aim, material and methods, results and conclusions;

- The aim of the study should end your introduction;

- Most of your introduction is part of material and methods;

- The tables you present should be concise and readable;

- Your conclusion should clearly answer your aim, present limitations and future implications.

Please rewrite your article regarding these comments and the journal's rules.

Author Response

Thank you for your valuable notes. We have taken them into consideration and revised the manuscript as follows:

  • Abstract should clearly state: introduction, aim, material and methods, results and conclusions;
    • Authors' response: We have added words: Background, Methods, Results and Conclusions into the abstract.
  • The aim of the study should end your introduction
    • Authors' response: We have moved the aim of the study in the end of introduction.
  • Most of your introduction is part of material and methods
    • Authors' response: We have revised and restructured the introduction and material.
  • The tables you present should be concise and readable
    • We have revised the tables and figures.  
  • Your conclusion should clearly answer your aim, present limitations and future implications.
    • Authors' response: We have revised the conclusion.

Please rewrite your article regarding these comments and the journal's rules.

Authors' response: We have reviewed and rewritten the manuscript as suggested.

Reviewer 2 Report

Thank you for the opportunity to review this manuscript.

In this study, the authors aimed to  develop and to test an instrument for measuring the humane behaviour management (HCMCB).

A convenience sample of health and social care professionals 21 (N=233) studying in University of Applied Sciences (N=13) was recruited. The EFA revealed a 14- 22 factor structure and included a total of 63 items. The Cronbach's alpha values for factors varied from  23

0.535 to 0.939. The participants rated their individual competence higher than leadership and o organizational culture. Finally, they concluded that HCMCB may be a useful tool for evaluating competences, leadership and organizational practices in the context of challenging behaviour. HCMCB should be further tested in 26 various international contexts involving challenging behaviour.

The introduction needs some changes; I think they repeat challenging behaviour many times in lines 31,32,35,3741,43,51,53,57,64. lines 31,32,35,3741,43,51,53,57,64.

The cited references are relevant to the research.

The research design is appropriate.

The results are presented, clearly, but The results are presented clearly, but there is a part regarding a low Cronbach's alpha value, and you explain that may be poor interrelatedness between items,  however the fact that in this study, KMO  values upper from 0.5 is appropriate based o extant literature as regards minimum threshold for the robustness of factor structure. It does not depend on sample size but on partial correlations, I think I understand, but I was hoping you could explain it better.

In addition, while the available data in this study support being psychometrically stable, some essential data are missing. For example, no measurements across extended periods could be used to assess sensitivity to change and long stability. Two test retests with n a 3-week interval will indicate that scores may vary over time in the same individuals exhibiting may consider fluctuating even within relatively short periods. Perhaps, I recommend it's necessary to add this or add it for future research.

The conclusions are supported by the results; the instrument should be tested The conclusion are supported by the results.

The instrument need should be tested in other research with large samples.

Best regards

Author Response

Thank you for your valuable comments. We have taken them into consideration and revised the manuscript as follows:

  • The introduction needs some changes; I think they repeat challenging behaviour many times in lines 31,32,35,3741,43,51,53,57,64.
    • We have reduced the use of "challenging behaviour" in introduction
  • The results are presented, clearly but there is a part regarding a low Cronbach's alpha value, and you explain that may be poor interrelatedness between items,  however the fact that in this study, KMO  values upper from 0.5 is appropriate based o extant literature as regards minimum threshold for the robustness of factor structure. It does not depend on sample size but on partial correlations, I think I understand, but I was hoping you could explain it better
    • Authors' response: As the KMO values were appropriate (<0.5), demonstrating a robust factor structure, a low Cronbach’s alpha value may be explained by poor interrelatedness between items or heterogeneous constructs, both of which signal for a need to revise or discard some of the items
  • In addition, while the available data in this study support being psychometrically stable, some essential data are missing. For example, no measurements across extended periods could be used to assess sensitivity to change and long stability. Two test retests with n a 3-week interval will indicate that scores may vary over time in the same individuals exhibiting may consider fluctuating even within relatively short periods. Perhaps, I recommend it's necessary to add this or add it for future research.
    • Authors' response: The reliability and validity of the instrument need more testing, for example assessing the sensitivity to change and longitudinal stability with a test-retest method.

  • The conclusions are supported by the results; The instrument need should be tested in other research with large samples.
    • Authors' response: We have revised the conclusions and taken this into consideration.

Reviewer 3 Report

Thank you very much for the opportunity to review this article. I was thrilled when I read the title of the work, but the content was disappointing. The method part is the weakest part.

please add to the abstract part how many items the latest version of the scale.

please add to figure 1 the references of the studies which you mentioned to get 155 items. 

Figure 1 and Table 1 are confusing I can not calculate 155 items.

Material and method: There are a lot of complicated parts. 

-When this data was collected? how long (minutes?) did it take to collect data?

-Research design: I think this is a methodological study. so please revise it.

-What are the inclusion criteria for participation in the study?

-authors mentioned that universe (N = 1685), but how do they calculate the sample size?

-What is the scale language? Finnish or English?

-I wonder how the participants were informed of the study and how they were protected from feeling coerced.  What happened to the students who declined to participate? 

I suggest that the authors rewrite the Discussion section. I'd like the results and discussion to be a bit stronger.  

I think the authors must make a more forceful statement in their conclusion.

Author Response

Thank you for your valuable comments. We have taken them in consideration and revised the manuscript carefully. 

  • please add to the abstract part how many items the latest version of the scale.
    • Authors' response: The number of items after EFA is mentioned in the abstract.
  • please add to figure 1 the references of the studies which you mentioned to get 155 items.
    • Authors' response: We have revised and clarified the Figure 1.
  • Figure 1 and Table 1 are confusing I can not calculate 155 items.
    • Authors' response: We have revised and clarified the Figure 1.
  • Material and method: There are a lot of complicated parts. -When this data was collected?
    • Authors' response: Data were collected in December 2020 – March 2021
  • how long (minutes?) did it take to collect data?
    • Authors' response: Completing the questionnaire took approximately 13 minutes (range 6-22).
  • Research design: I think this is a methodological study. so please revise it.
    • Authors' response: Research design was found appropriate by other reviewers. 
  • What are the inclusion criteria for participation in the study?
  • Authors' response: Please see chapter 2.5.1: Therefore, a participant had to be a health or social care professional (e.g., Registered Nurse, Occupational Therapist or Bachelor of Social Services) with a Bachelors’ degree who was studying in a Masters’ degree Program at a university of applied sciences. (lines 202-203)
  • authors mentioned that universe (N = 1685), but how do they calculate the sample size?
    • Authors' response: Appropriate sample size when conducting EFA is five to ten subjects per item (line 200)
  • What is the scale language? Finnish or English?
    • Authors' response: The instrument is in Finnish language (Figure 1)
  • I wonder how the participants were informed of the study and how they were protected from feeling coerced.  What happened to the students who declined to participate? 
    • Authors' response: Participants were provided with information about the study via email, including an explanation that their participation was voluntary. Participation or withdrawal were not controlled by universities or the researcher and had no influence on students’ ongoing studies or study assessments. (chapter 2.5.4)
  • I suggest that the authors rewrite the Discussion section. I'd like the results and discussion to be a bit stronger.  
    • Authors' response: We have revised and rewritten the Discussion.
  • I think the authors must make a more forceful statement in their conclusion.
    • Authors' response: We have revised and rewritten the Discussion.

Round 2

Reviewer 1 Report

Dear authors:

Congrats for improving your work.

Reviewer 3 Report

Dear authors, thank you so much for your effort. Good job, regards.